# P-Flow: A Fast and Data-Efficient Zero-Shot TTS through Speech Prompting

**Sungwon Kim**[1,2*†] **Kevin J Shih**[1]**, Rohan Badlani**[1]**, João Felipe Santos**[1]**, Evelina Bhakturina**[1]**,
**Mikyas Desta**[1]**, Rafael Valle**[1†]**, Sungroh Yoon**[2,3†]**, Bryan Catanzaro**[1]

NVIDIA[1]
Department of Electrical and Computer Engineering, Seoul National University[2]
Interdisciplinary Program in Artificial Intelligence, Seoul National University[3]

## Abstract

While recent large-scale neural codec language models have shown significant improvement in zero-shot TTS by training on thousands of hours of data, they suffer from drawbacks such as a lack of robustness, slow sampling speed similar to previous autoregressive TTS methods, and reliance on pre-trained neural codec representations. Our work proposes P-Flow, a fast and data-efficient zero-shot TTS model that uses speech prompts for speaker adaptation. P-Flow comprises a speech-prompted text encoder for speaker adaptation and a flow matching generative decoder for high-quality and fast speech synthesis. Our speech-prompted text encoder uses speech prompts and text input to generate speaker-conditional text representation. The flow matching generative decoder uses the speaker-conditional output to synthesize high-quality personalized speech significantly faster than in real-time. Unlike the neural codec language models, we specifically train P-Flow on LibriTTS dataset using a continuous mel-representation. Through our training method using continuous speech prompts, P-Flow matches the speaker similarity performance of the large-scale zero-shot TTS models with two orders of magnitude less training data and has more than $20\times$ faster sampling speed. Our results show that P-Flow has better pronunciation and is preferred in human likeness and speaker similarity to its recent state-of-the-art counterparts, thus defining P-Flow as an attractive and desirable alternative. We provide audio samples on our demo page.

## 1 Introduction

Zero-shot TTS (Text-to-Speech) refers to the goal of generating text-conditioned speech with novel speaker characteristics at inference without the need for additional training. The speaker characteristics are provided by a short reference audio segment, which in recent works can be as short as 3 seconds long [37, 19, 6]. Many recent advances have come at a cost: larger datasets, more complicated training setups, additional quantization steps, additional pretraining tasks, and computationally expensive autoregressive formulations. Our work explores whether a more efficient solution exists. By staying within the general framework of prompt continuation, as often seen in recent large language model works, we demonstrate that similar results can be achieved with simpler training pipelines, significantly less data, and faster inference.

Following recent successes in the zero-shot and few-shot capabilities of large scale language models such as GPT [2], it has been generally accepted that an analogous approach will yield similar results

---

\* Work done as a research intern at NVIDIA.

† Corresponding authors: Sungwon Kim: ksw0306@snu.ac.kr, Rafael Valle: rafaelvalle@nvidia.com, Sungroh Yoon: sryoon@snu.ac.kr

37th Conference on Neural Information Processing Systems (NeurIPS 2023).

in speech synthesis. However, it is not yet clear that recent approaches in large scale representation learning have a significant advantage over existing and cheaper traditional representations such as mel-spectrograms. Of note, recent works such as SPEAR-TTS and VALL-E make use of pretrained quantized neural audio codec models such as Encodec and SoundStream [10, 40]. The costs of using such a model are expensive. Feature extraction is significantly more computationally demanding than extracting mel-spectrograms from audio. Encodec in particular contains autoregressive compontents in both the encoding and decoding, which may be an issue where low-latency is necessary. Further, having too many different versions of such a codec inhibit interoperability across models. While it is likely that neural codecs may yet have significant advantages in the near future, our work demonstrates competitive results with standard mel-spectrograms, suggesting that the costs of neural codecs in the zero-shot TTS setting is not yet justified by any significant improvements in generalization.

Improving inference speed is the other focus of this work. Our work improves upon the latency of prior approaches by reducing reliance on autoregressive formulations, and by employing recent advances in ODE-based generative modeling such as Flow Matching [23, 24, 35]. In addition to autoregressive components within the neural codec architectures, prior works such as SPEAR-TTS and VALL-E also employ autoregressive decoders to infer audio token sequences. While autoregressive modeling is a natural choice for sequential data such as audio signals, our work demonstrates that a lower-latency non-autoregressive formulation is both viable and competitive. Another common computational bottleneck is the incorporation of probabilistic generative models such as denoising diffusion models in recent works. Such architectures exist in many recent state-of-the-art speech and audio synthesis models, but come at the cost of requiring numerous function evaluations during inference. Distillation techniques exist to reduce the number of function evaluations, but require additional training stages, further complicating the training pipeline. In contrast, we experiment with recently proposed flow-matching algorithms. These models are closely related to diffusion models, but actively encourage simpler and straighter trajectories, which in turn require significantly fewer function evaluations during inference – all without the need for additional distillation stages.

Our work proposes P-Flow, high-quality, data-efficient, and fast zero-shot TTS model with emergent in-context learning capabilities for speaker adaptation. Inspired by recent success using language-model prompting to achieve zero-shot TTS results, we introduce a speech-prompted text encoder within a non-autoregressive TTS model. P-Flow combines this prompted text encoder with a flow-matching generative decoder to efficiently sample high-quality mel-spectrograms. Our speech-prompted text encoder incorporates a 3-second random segment of speech as a prompt along with the text input and learns to predict the mel-spectrogram using the speech-prompted text input. Given the speaker-conditional output from the text encoder, our flow-matching decoder is trained to predict a vector field that models the probabilistic distribution of mel-spectrograms conditioned on the text encoder's output. Our prompting approach directly utilizes random segments of speech as input, which poses a risk of P-Flow learning trivial solutions such as the identity function for the random segment. We address the potential problem by masking the loss for the random segment during training. Our contributions are as follows:

- We propose a speech prompt approach for the non-autoregressive zero-shot TTS model which surpasses the speaker embedding approach and provides in-context learning capabilities for speaker adaptation.
- We propose a flow matching generative model for a high-quality and fast zero-shot TTS that significantly improves the synthesis speed and sample quality compared to the large-scale autoregressive baseline.
- We demonstrate comparable speaker adaptation performance to the large-scale autoregressive baseline using significantly fewer training data and a small transformer-based encoder, highlighting the effectiveness of the proposed speech prompting approach.
- P-Flow achieves an average inference latency of 0.11 seconds on an NVIDIA A100 GPU.

## 2   Related work

**Zero-Shot Speaker Adaptive TTS (Zero-Shot TTS):**  Instead of specifying speaker identity with discrete embedding vectors as in the case of multi-speaker TTS models, zero-shot TTS formulations assume the ability to extract speaker embedding equivalents on-the-fly from short audio samples. Common approaches either use a separate audio-sample-to-embedding encoder to extract speaker

vectors on the fly [17, 27, 20, 7, 3, 38], or formulate the problem as a continuation task with language model architectures and probabilistic generative models. Methods such as VALL-E and SPEAR-TTS [37, 19] employ the prompting paradigm from large transformer-based language models. Notably, they use the reference audio as the prompt, from which the model is expected to decode the most likely continuation. The decoding procedure is further conditioned with the provided text transcript to achieve a text-to-speech formulation. Relatedly, approaches such as [28, 9, 25] use the data inpainting paradigm from generative probabilistic models, treating the speech-to-be-synthesized as the missing data to continue the available data (prompted audio). Just as in the LM-based approach, the inpainting is guided by conditioning on the target text sequence. Our work operates in the same prompt-based zero-shot TTS setting as VALL-E and SPEAR-TTS, establishing competitive results with much less data, and additionally providing faster inference speeds via flow-matching-based generative models and non-autoregressive formulations.

Perhaps most architecturally related to our work is $A^3T$ [4], which similarly trains a transformer ($A^3T$ uses a Conformer [12]) model to infer text-conditional mel spectrograms with a masked reconstruction loss. However, $A^3T$ uses a forced aligner, requires a text transcript for the zero-shot audio prompt, and focuses primarily on the pretraining task and audio reconstruction, with limited results in the zero-shot TTS domain. In contrast, our proposed approach uses an end-to-end audio-text alignment mechanism, uses short 3 second audio-only prompt, and focuses primarily on the zero-shot TTS task.

**Generative Sampling with Fewer Steps:** Generative models in the score matching and denoising diffusion model family typically require hundreds of iterative steps. It is common to reduce the number of steps required via approaches such as the DDIM [32] framework and distillation [31]. Notably, distillation will progressively map every two steps to a single step, halving the number of total steps required with every iteration of the algorithm. Recent works such as flow matching [24, 23, 35] and consistency training (from scratch) [33] offer a single-stage approach, encouraging approximately straight, causal trajectories in the initial model, thereby reducing the need for further distillation stages. Our work serves as a demonstration of the use of such approaches as flow matching in large-scale practical audio applications.

# 3 Method

Our work aims to provide in-context learning capabilities for zero-shot speaker adaptation in a high-quality and fast non-autoregressive TTS model. To avoid the potential bottleneck of speaker embedding approaches to extract speaker information from the reference speech, we adopt a prompting approach that directly utilizes the reference speech as a prompt for speaker adaptation, similar to neural codec language models. In addition to a speech prompting method, we use a flow matching generative model as our decoder for efficient sampling. We provide an overview of our method in Section 3.1, followed by a detailed explanation of our decoder in Section 3.2, and we describe the remaining details in Section 3.3.

## 3.1 P-Flow

P-Flow training is similar to that of masked-autoencoders. Given <text, speech> paired data, we denote the mel-spectrogram of the speech as $x$, and the text as $c$. Let $m^p$ be a indicator mask on the sequence $x$, masking out a randomly positioned 3 second segment from $x$. We define $x^p = (1-m^p) \cdot x$. We use a $p$ superscript here indicating that this variable will be replaced with arbitrary 3-second prompts during zero-shot inference. The training objective for P-Flow is as follows: reconstruct the $x$ given $c$ and some segment $x^p$. Note that even though we expect P-Flow to reconstruct the entirety of $x$ during training, including the provided $x^p$, *the model does the exact positioning of $x^p$ within $x$.*

P-Flow learns to model the conditional probability distribution of speech $p(x|c, x^p)$. Unlike many zero-shot TTS models that compress all the information in the speech into a fixed-size vector, we introduce a text encoder $f_{enc}$, which takes the text $c$ and the random segment $x^p$ as a speech prompt to generate a speaker-conditional text representation. We refer to our text encoder as a speech-prompted text encoder. The output of this text-encoder is then mapped to mel-spectrograms using a flow-matching generative model, and finally to waveform using an audio vocoder.

**Speech-Prompted Text Encoder:** As shown in Fig. 1a, we input the mel-spectrogram of a random segment $x^p$ as a speech prompt along with the text input $c$. We then project both to the same

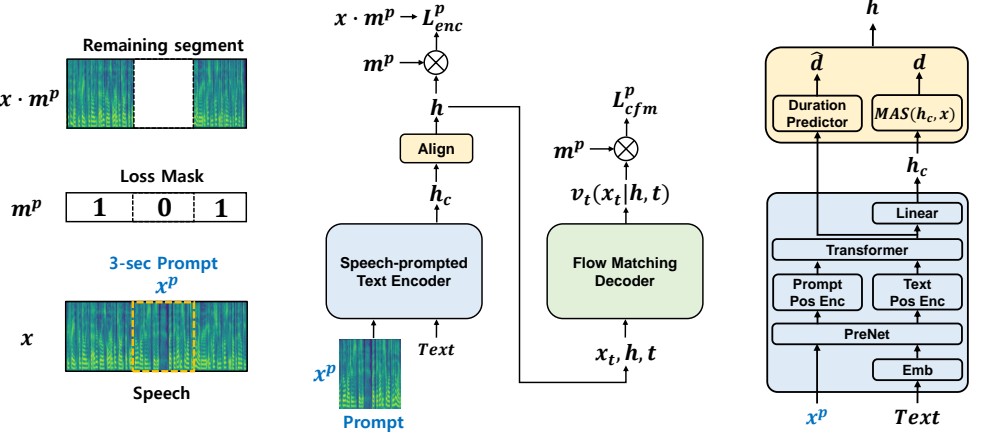

(a) An overview of P-Flow          (b) Speech-prompted Text Encoder

Figure 1: **The overall diagram of P-Flow.** P-Flow is a model composed of a speech-prompted text encoder, which outputs a representation containing speaker information from the speech prompt, and a flow matching decoder generates high-quality speech significantly faster than real-time.

dimensions to use as inputs to the text encoder. The role of the speech-prompted text encoder is to generate a speaker-conditional text representation $h_c = f_{enc}(x^p, c)$ using the speaker information extracted from the prompt $x^p$. Similar to large-scale codec language models, we employ a non-autoregressive transformer architecture that can attend to speech prompts at arbitrary text positions.

To train the speech-prompted text encoder to effectively extract speaker information from the speech prompt, we use an encoder loss that directly minimizes the distance between the text encoder representation and the mel-spectrogram. In addition to its original purpose in Grad-TTS [30] for reducing sampling steps in a diffusion-based single-speaker TTS model, it also encourages the encoder to incorporate speaker-related details into the generated text representation.

As speech-prompted encoder output $h_c$ and the mel-spectrogram $x$ have different lengths, we align the text encoder output with the mel-spectrogram using the monotonic alignment search (MAS) algorithm proposed in Glow-TTS [21]. By applying MAS, we derive an alignment $A = MAS(h_c, x)$ that minimizes the overall L2 distance between the aligned text encoder output and the mel-spectrogram. Based on the alignment $A$, we determine the duration $d$ for each text token and expand the encoder output $h_c$ by duplicating encoder representations according to the duration of each text token. This alignment process results in text encoder output $h$ that aligns with mel-spectrogram $x$. The fairly straightforward reconstruction loss is written as $L_{enc} = MSE(h, x)$.

In practice, even though the model is not given the exact positioning of $x^p$ within $x$ during training, we found the model to still collapse to a trivial copy-pasting of $x^p$. To avoid this, we simply mask out the reconstruction loss for the segment corresponding to $x^p$. Despite this, the final model is still capable of inferring a continuous mel-spectrogram sequence. We define the masked encoder loss $L_{enc}^p$ by using $m^p$ for the random segment $x^p$ in the mel-spectrogram $x$:

$$L_{enc}^p = MSE(h \cdot m^p, x \cdot m^p) \tag{1}$$

By minimizing this loss, the encoder is trained to extract speaker information as much as possible from the speech prompt in order to generate an aligned output $h$ that closely resembles the given speech $x$, which results in enhancing in-context capabilities for speaker adaptation.

**Flow Matching Decoders:** To perform high-quality and fast zero-shot TTS, we use a flow-matching generative model as the decoder for modeling the probability distribution $p(x|c, x^p) = p(x|h)$. Our flow-matching decoder models the conditional vector field $v_t(\cdot|h)$ of Continuous Normalizing Flows (CNF), which represents the conditional mapping from standard normal distribution to data distribution. The decoder is trained using a flow-matching loss $L_{cfm}^p$ that also applies a mask $m^p$ for the random segment as in $L_{enc}^p$. More details will be provided in Section 3.2.

**Duration Predictor:** To reproduce text-token durations during inference where MAS is unavailable, we use a duration predictor trained in a manner similar to [21] . We use the hidden representation

of the speech-prompted text encoder as its input without additional speaker conditioning, given this representation already contains speaker information. It is trained simultaneously with the rest of the model with detached inputs to avoid affecting the text-encoder training. The duration predictor estimates the log-scale duration $\log \hat{d}$ for each text token, and the training objective for the duration predictor $L_{dur}$ is to minimize the mean squared error with respect to log-scale duration $\log d$ obtained via MAS during training.

The overall training loss for P-Flow is $L = L_{enc}^p + L_{cfm}^p + L_{dur}$. Zero-shot inference uses a random chunk from a reference sample as the speech prompt along with the desired transcript as inputs to the speech-prompted text encoder to obtain $h_c$. We do not provide a transcript corresponding to the speech prompt. Then, using the predicted duration $\hat{d}$ from the duration predictor, we expand the encoder output $h_c$ to obtain $h$ and generate the personalized speech using the flow-matching decoder.

### 3.2 Flow Matching Decoder

We use Flow Matching to model the mel-spectrogram decoder task's conditional distribution: $p(x|h)$. We first provide a brief overview of flow matching, followed by describing our sampling procedure and additional qualitative improvements through a guidance-related technique.

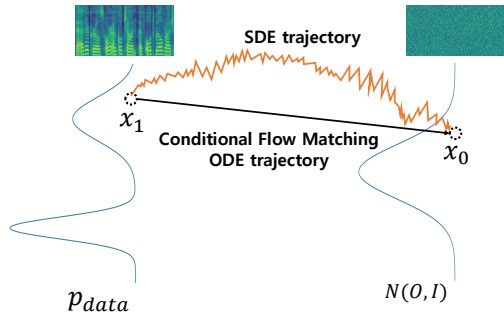

Figure 2: Conditional Flow Matching [23] can be supervised with linear conditional trajectories during training when mapping between the complex $p_{\text{data}}$ and the sampling Gaussian, leading to simpler marginal trajectories during inference. Simpler trajectories require fewer function evaluations.

**Flow Matching Overview:** Flow Matching [23, 35, 24] is a method for fitting to the time-dependent probability path between our data density $p_1(x)$ and our simpler sampling density $p_0(x)$ (assumed to be the standard normal). It is closely related to Continuous Normalizing Flows, but is trained much more efficiently in a simulation-free fashion, much like the typical setup for Diffusion and Score Matching models [13, 16, 34]. We adopt Conditional Flow Matching as specified in [23] as their formulation encourages simpler and often straighter

trajectories between source and target distributions. Simpler trajectories allow for test-time sampling in fewer steps without the need for additional distillation. We will ignore the conditional variable $h$ for notational simplicity in this overview.

Following Lipman et al. [23], we define the flow $\phi : [0, 1] \times \mathbb{R}^d \to \mathbb{R}^d$ as the mapping between our two density functions using the following ODE:

$$\frac{d}{dt}\phi_t(x) = v_t(\phi_t(x)); \quad \phi_0(x) = x \tag{2}$$

Here, $v_t(x)$ is the time-dependent vector field and specifying the trajectory of the probability flow through time. $v_t(x)$ is also our learnable component, henceforth denoted as $v_t(\phi(x); \theta)$. To sample from the distribution, we sample from the sampling distribution $p_0$ as our initial condition at $t = 0$ and solve the ODE in Eq. (2). Notably, the formulation [23] encourages straighter trajectories, ultimately allowing us to cheaply approximate the ODE solution with 10 Euler steps with minimal loss in quality.

It turns out that determining the marginal flow $\phi_t(x)$ is difficult in practice. Lipman thus formulates it as marginalizing over multiple conditional flows $\phi_{t,x_1}(x)$ as follows:

$$\phi_{t,x_1}(x) = \sigma_t(x_1)x + \mu_t(x_1) \tag{3}$$

Here, $\sigma_t(x_1)$ and $\mu_t(x_1)$ are time-conditional affine transformations for parameterization the transformation between Gaussian distributions $p_1$ and $p_0$. Finally, let $q(x_1)$ be the true but likely non-Gaussian distribution over our data. We define $p_1$ as a mixture-of-Gaussian approximation of $q$ by perturbing individual samples with small amounts of white noise with $\sigma_{\min}$ (empirically set to $0.01$). We can specify our trajectories without complications from stochasticity as in SDE formulations.

Taking advantage of this, Lipman et al. recommend simple linear trajectories, yielding the following parameterization for $\phi_t$:

$$\mu_t(x) = tx_1, \; \sigma_t(x) = 1 - (1 - \sigma_{\min})t \tag{4}$$

Training the vector field is performed using the conditional flow matching objective function:

$$L_{CFM}(\theta) = \mathbb{E}_{t \sim U[0,1], x_1 \sim q(x_1), x_0 \sim p(x_0)} \|v_t(\phi_{t,x_1}(x_0); \theta) - \frac{d}{dt}\phi_{t,x_1}(x_0)\|^2 \tag{5}$$

Plugging Eq. (4) in to Eq. (3) and (5), we get our final CFM objective:

$$L_{CFM}(\theta) = \mathbb{E}_{t,q(x_1),p(x_0)} \|v_t(\phi_{t,x_1}(x_0); \theta) - (x_1 - (1 - \sigma_{\min})x_0)\|^2 \tag{6}$$

**Masked Flow Matching Loss:** Recall that our flow matching decoder models the distribution $p(x|h)$. Because $h$ is the output of the text encoder which was provided by the subsegment $x^p \in x$, we found it again necessary to mask out the loss for parts of the output corresponding to $x^p$ to prevent trivial solutions. Let the generic $v_t(x_t; \theta)$ be parameterized in our setup as $\hat{v}_\theta(x_t, h, t)$ to account for the conditional. Here, $t$ is represented with a continuous sinusoidal embedding. This gives us the masked CFM objective:

$$L_{CFM}^p(\theta) = \mathbb{E}_{t,q(x_1),p(x_0)} \|m^p \cdot (\hat{v}_\theta(\phi_{t,x_1}(x_0), h, t) - (x_1 - (1 - \sigma_{\min})x_0))\|^2 \tag{7}$$

**Sampling:** The conditional flow matching loss marginalizes over conditional vector fields to achieve the marginal vector field, the latter of which is used during sampling. While the linearly interpolated conditional trajectories as specified in Eq. (4) do not guarantee the same degree of straightness in the resulting marginal, we still get something fairly close. Within the context of this work, we found the conditional flow matching formulation to result in simple enough trajectories such that it is sufficient to use the first-order Euler's method with around 10 steps to solve the ODE during inference. Sampling with $N$ Euler steps is performed with the following recurrence relation:

$$x_0 \sim \mathcal{N}(0, I); \;\; x_{t+\frac{1}{N}} = x_t + \frac{1}{N}\hat{v}_\theta(x_t, h, t) \tag{8}$$

**Guided Sampling:** We find that pronunciation clarity can be further enhanced by applying techniques from a classifier-free guidance method [14]. In a related work, GLIDE [29] amplifies their text-conditional sampling trajectory by subtracting the trajectory for an empty text sequence. We employ a similar formulation, guiding our sampling trajectory away from the average feature vector computed from $h$, denoted as $\bar{h}$. $\bar{h}$ is computed by averaging the expanded representation $h$ along the time axis to obtain a fixed-size vector and then duplicated along the time axis. Let $\gamma$ be our guidance scale. Our guidance-amplified Euler formulation is as follows:

$$x_{t+\frac{1}{N}} = x_t + \frac{1}{N}(\hat{v}_\theta(x_t, h, t) + \gamma(\hat{v}_\theta(x_t, h, t) - \hat{v}_\theta(x_t, \bar{h}, t))) \tag{9}$$

### 3.3 Model Details

The high-level model architecture is shown in Fig. 1. Our model comprises 3 main components: the prompt-based text encoder, a duration predictor to recover phoneme durations during inference, and a Wavenet-based flow matching decoder. Experiments demonstrate strong zero-shot results despite our text-encoder comprising only a small transformer architecture of 3M parameters. We provide additional architectural details for each component in Section B.

## 4 Experiments

**Training and Inference Settings:** P-Flow is trained on a single NVIDIA A100 GPU for 800K iterations, using a batch size of 64. We utilize the AdamW optimizer [26] with a learning rate of 0.0001. A G2P model [5] preprocesses the text into the International Phonetic Alphabet (IPA) format. During inference, we generate mel-spectrograms using 10 Euler steps in the flow matching decoder with a guidance scale of 1. Mel-spectrogram to waveform is performed using the pre-trained universal Hifi-GAN, available in the HiFi-GAN repository[3]. To be compatible with Hifi-GAN, our audio representation is 22kHz audio represented with an 80-bin mel-spectrogram, with FFT parameters:

---

[3]Hifi-GAN: https://github.com/jik876/hifi-gan

window size 1024 and hop length 256. In this setup, a 3-second mel-spectrogram is fed as a speech prompt with a length of $\lceil \frac{3 * 22050}{256} \rceil = 259$.

**Data:** We train P-Flow on LibriTTS [41]. LibriTTS training set consists of 580 hours of data from 2,456 speakers. We specifically use data that is longer than 3 seconds for speech prompting, yielding a 256 hours subset. For evaluation, we follow the experiments in [37, 19] and use LibriSpeech test-clean, assuring no overlap exists with our training data. We resample all datasets to 22kHz.

**Evaluation:** We compare P-Flow with two zero-shot speaker-adaptive TTS models, YourTTS [7] and VALL-E [37]. YourTTS is a VITS-based multi-speaker TTS, which performs zero-shot speaker adaptation through speaker embedding extracted from a pre-trained speaker encoder. VALL-E achieves zero-shot speaker adaptation through prompting, resulting in much higher speaker similarity than YourTTS. We compare our model with the two models in the same way as described in [37], and the objective metrics for each model were directly taken from [37].

Replicating the experiment in [37, 19], we evaluate samples ranging from 4 to 10 seconds from the LibriSpeech test-clean dataset, resulting in a total of 2.2 hours of data. For each paired data $(x_i, c_i)$, we extract a 3-second reference speech $x_j^p$ from a different sample $x_j$ spoken by the same speaker and generate the synthesized speech $x_i^{gen}$ for the text $c_i$.

**Objective Metrics:** We measure the inference latency and two objective metrics from [37], word error rate (WER), and speaker embedding cosine similarity (SECS). Inference latency of P-Flow and the autoregressive baseline, VALL-E are evaluated on an NVIDIA A100 GPU. To compute inference latency for P-Flow, we average the text-to-mel generation time for all the samples used in our evaluation for LibriSpeech. Because VALL-E is not open-sourced, we use a PyTorch transformer[4] of the same size as used in [37] and feed random tensors repeatedly corresponding to the duration of $x_i$ to obtain a proxy for the inference latency.[5]

We evaluate pronunciation accuracy with WER, using the HuBERT [15] ASR model to measure transcription errors from generated samples. Speaker similarity is measured with SECS between the generated sample $x_i^{gen}$ and the re-synthesized output of the 3-second reference speech $x_j^p$ as in [37]. We compute the speaker embedding using the pre-trained WavLM-TDNN [8] used in [37]. We also measure the WER and SECS for the re-synthesized output of the ground truth audio $x_i$ as an upper bound, reported as "GT (HIFI-GAN)" in Table 2. Note that the final mapping to waveform is different between models: P-Flow uses Hifi-GAN [22] while VALL-E uses EnCodec [11], necessarily influencing automatic metrics. We partially address this with our user study setup for speaker similarity.

**User Study:** We measure user preference score between VALL-E and P-Flow. Qualified evaluators must first pass hearing test where they count the number of short sinusoidal audio segments within an audio clip. We obtain evaluation scores for 8 samples from each evaluator with a minimum of 30 evaluators per comparison experiment. We evaluate the preference for naturalness, acoustic quality, and human likeness using comparative mean opinion score (CMOS). Preference for speaker similarity is reported using comparative speaker similarity mean opinion score (SMOS). SMOS evaluators are provided with a 3-second reference audio, which is the ground truth audio rather than re-synthesized audio to avoid being influenced by vocoder artifacts. Each preference score ranges from -2 to 2, using pairwise comparisons between the LibriSpeech samples from the VALL-E demo page.

### 4.1 Prompting v.s. Separate Speaker Encoder

This ablation study demonstrates the effectiveness of speaker conditioning through speech prompting. For the baseline, instead of directly inputting the speech prompt into the text encoder, we encode a random segment of speech $x^p$ to a fixed-size speaker embedding using a speaker encoder with the same architecture as the text encoder. The speaker embedding is obtained by averaging the last hidden representations of the speaker encoder along the time axis and normalizing it. This speaker embedding is concatenated with the text input of the text encoder. The speaker encoder is jointly trained with the model and follows the same training objective, except for excluding the loss mask

---

[4]https://pytorch.org/docs/stable/generated/torch.nn.TransformerEncoder.html

[5]Following [37], we measure inference on a 12-layer transformer with a hidden size of 1024 and 16 heads. Given that VALL-E's neural audio codec representation is at 75Hz, we approximate the inference latency of VALL-E by measuring the latency required for $75 * n + 7$ forward passes to generate $n$ seconds of audio.

Table 1: Objective metrics for ablation study. "P-Flow (w/o Prompt)" refers to the baseline conditioning on a fixed-size speaker embedding.

| MODEL | WER↓ | SECS↑ |
|---|---|---|
| GT (HIFI-GAN) | 2.4 | 0.64 |
| P-FLOW (W/O PROMPT) | 2.9 | 0.373 |
| P-FLOW | **2.6** | **0.544** |

for the speech prompt. For evaluation, we also sample using 10 Euler steps with a guidance scale of 1, similar to P-Flow.

In Table 1, we compare the objective metrics of P-Flow and the baseline for the fixed-size speaker embedding approach. Despite having similar WER values, it can be observed that introducing speech prompt-based speaker conditioning significantly enhances SECS compared to the baseline. This confirms that even without changing other aspects, the use of speech prompts leads to a substantial enhancement in speaker similarity. Through this ablation study, we demonstrate that the speech prompt-based approach in P-Flow, which utilizes the speech prompt as direct input and adapts it through the self-attention mechanism to optimize the loss for the remaining segment, is more effective for zero-shot TTS compared to encoding all the speaker information into a fixed-size vector.

Table 2: The amount of training data and objective metrics for zero-shot TTS models. The SECS of "GT (HIFI-GAN)" is computed between the re-synthesized output of 3-second reference speech and that of ground truth. WER and SECS for the baselines are taken from [37]. The inference latency for VALL-E is approximated using the same size of transformer architecture, as mentioned in Section 4.

| MODEL | DATA (HOURS) | WER↓ | SECS↑ | INFERENCE LATENCY(S)↓ |
|---|---|---|---|---|
| GT (HIFI-GAN) | | 2.4 | 0.64 | |
| YOURTTS[†] | 500+ | 7.7 | 0.337 | |
| VALL-E[†] | 60,000 | 5.9 | **0.580** | $2.515 \pm 0.040$ |
| VALL-E CONTINUAL[†] | 60,000 | 3.8 | 0.508 | $2.515 \pm 0.040$ |
| P-FLOW (PROPOSED) | **260** | **2.6** | 0.544 | $\mathbf{0.115 \pm 0.004}$ |

## 4.2 Model Comparison

Samples for each model can be accessed through the demo page.[6] We highly encourage reviewers to listen to the samples.

**Pronunciation Accuracy and Sample Quality:** Our results in Table 2 show that P-Flow offers similar WER to GT (HIFI-GAN) and demonstrates significantly better pronunciation accuracy compared to other baselines while being trained on two orders of magnitude less data. This indicates that with training on the more accurately transcribed LibriTTS dataset, we can achieve pronunciation accuracy close to the ground truth.

Table 3: Zero-Shot Speech Synthesis Subjective Metrics. Preference scores $\geq 0$ indicate preference for P-Flow.

| MODEL | CMOS↑ | SMOS↑ |
|---|---|---|
| P-FLOW VS VALL-E | $\mathbf{0.27 \pm 0.10}$ | $\mathbf{0.23 \pm 0.13}$ |

**Speaker Similarity Metrics:** Compared to the previous speaker embedding-based zero-shot TTS model, YourTTS, P-Flow demonstrates more effective speaker adaptation through speech prompting using a similar amount of training data. Considering our model text-encoder only has 3M parameters, competitive performance with large-scale autoregressive baselines suggests achieving high SECS does not require tens of thousands of hours of audio data, quantization, neural codec models, or

---

[6] Demo: https://research.nvidia.com/labs/adlr/projects/pflow

large autoregressive models. Further, combining our speaker conditioning ablation study with these results and the fact that our decoder does not have any additional speaker conditioning beyond what is provided by the text encoder, we believe that prompting is one of the most important factors in obtaining high SECS. Finally, we see higher preference scores for P-Flow than for previous SOTA zero-shot models in the human evaluation of speaker similarity with the ground truth audio in Table 3. This indicates that our speech-prompted text encoder enhances the in-context learning capabilities, similar to large-scale autoregressive transformers.

**CMOS:** In Table 3, we can observe the CMOS results of pairwise comparisons between samples from VALL-E and P-Flow, evaluating naturalness, prosody, and audio quality. Table 3 demonstrates that P-Flow exhibits significantly better sample quality compared to VALL-E. This CMOS result shows that generative modeling using continuous representation, which has been widely used in TTS, exhibits superior performance compared to generation using discrete tokens. We attribute this result not only to the performance difference between the models but also to the additional performance degradation introduced during the encoding-decoding process using neural codec representation. It highlights the advantages of continuous representation and showcases its effectiveness.

**Inference Latency:** Defaulting to 10 Euler steps for P-Flow's decoder, we measure inference latency for audio samples from LibriSpeech with an average length of 5.6 seconds. Tables 2 and 3 show our model achieves an inference latency of 0.1 seconds, more than 20 times faster than VALL-E, while maintaining high speaker similarity, sample quality, and pronunciation accuracy. While the autoregressive baseline has a generation time proportional to the duration of the audio, the non-autoregressive P-Flow maintains nearly constant latency regardless of the length. A 20-second audio sample would take P-Flow 0.12 seconds in our experimental setup.

Table 4: Effect of guidance scale $\gamma$ and Euler steps $N$ for our flow matching generative decoder. Our experiments find that automatic metric numbers plateau quickly after 5 steps, but we continue to hear qualitative improvements up to 10. Significant improvements are also seen due to the guidance.

| MODEL | $\gamma$ | $N$ | WER↓ | SECS↑ | INFERENCE LATENCY(S)↓ |
|---|---|---|---|---|---|
| P-FLOW (DEFAULT) | 1 | 10 | 2.6 | 0.544 | $0.115 \pm 0.004$ |
| P-FLOW | 0 | 10 | 3.7 | 0.492 | $0.115 \pm 0.004$ |
| P-FLOW | 2 | 10 | 2.6 | 0.546 | $0.115 \pm 0.004$ |
| P-FLOW | 1 | 1 | 2.7 | 0.420 | $0.028 \pm 0.004$ |
| P-FLOW | 1 | 2 | 2.9 | 0.522 | $0.037 \pm 0.004$ |
| P-FLOW | 1 | 5 | 2.6 | 0.549 | $0.067 \pm 0.004$ |
| P-FLOW | 1 | 20 | 2.7 | 0.540 | $0.210 \pm 0.005$ |

## 4.3 Effect of Guidance Scale and Euler Steps

Table 4 demonstrates how the guidance scale $\gamma$ and number of Euler steps $N$ affects objective metrics. As shown in Table 2 and 4, P-Flow outperforms YourTTS even without the guidance method, showing the effectiveness of the speech-prompting method for speaker adaptation. Applying the guidance method improves pronunciation accuracy and brings our SECS metric closer to that of the state-of-the-art zero-shot TTS model without explicit training for guidance. These results confirm that the guidance method applied to P-Flow further boosts the effectiveness of speech prompting. Both guidance scales $\gamma = \{1, 2\}$ perform well. (We default to $\gamma = 1$)

Regarding the Euler steps, P-Flow demonstrates low WER even with very few Euler steps, indicating that the flow matching generative model generates speech that can be accurately recognized by the ASR model regardless of audio quality. We default to 10 Euler steps for external comparisons because SECS values plateau roughly after 5 Euler steps and we notice that good audio quality is achieved at around 10 steps. In Table 4, the ASR metrics and SECS metrics do not directly represent the acoustic quality of the samples according to the Euler steps. Therefore, we provide mean opinion scores (MOS) for the acoustic quality based according to the Euler steps in Section A.1. We further provide audio samples corresponding to our default inference settings in our demo page.

# 5 Discussion and Limitations

P-Flow is a fast and data-efficient flow matching model for zero-shot TTS that achieves comparable naturalness and speaker adaptation performance to its large-scale and autoregressive counterparts. The core of the approach is the use of the prompt-continuation setup in recent LLM applications while avoiding complex and expensive setups from recent works in zero-shot TTS. In some ways, our work serves as a strong yet simple baseline for future approaches, as we demonstrate that state of the art performance in this task can be achieved without super large-scale datasets, complex training setups, representation quantization steps, pretraining tasks, and expensive autoregressive formulations.

The focus of this work is primarily on zero-shot capabilities with respect to text-encoding and audio decoding, whereas zero-shot capabilities of the duration predictor remain limited and the subject of future work. Furthermore, high-quality zero-shot TTS can yield negative social impact through applications such voice impersonation of public figures and non-consenting individuals, which we raise awareness for as a potential misuse of this technology.

## Acknowledgements

This work was partially supported by the BK21 FOUR program of the Education and Research Program for Future ICT Pioneers, Seoul National University in 2023 and Institute of Information & Communications Technology Planning & Evaluation (IITP) grant funded by the Korea government (MSIT) [NO.2021-0-01343, Artificial Intelligence Graduate School Program (Seoul National University)].

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

# A  Additional Results

## A.1  Effect of Euler Steps for Acoustic Quality

We present the objective metrics according to the Euler steps in the result section of the main paper. Since these objective metrics have limitations in representing acoustic quality with respect to the Euler step, we also evaluate the sample quality based on the Euler steps and provide it as an additional metric. We measure the acoustic quality using 5-scale Mean Opinion Scores (MOS). We inquire each human evaluator to assess the acoustic quality of each sample, and we evaluate it with the participation of more than 50 human evaluators.

Table 5 presents MOS along with SECS and inference latency shown in the results section, based on the Euler step $N$. The table demonstrates that as the number of Euler steps increases, the acoustic sample quality improves. We choose Euler step 10 as the default, as it ensures a high speaker similarity while providing a good balance between inference latency and sample quality.

Table 5: Mean Opinion Scores (MOS) for the acoustic quality and Objective Metrics according to the Euler steps $N$.

| Model | $N$ | MOS↑ | SECS | Inference Latency(s)↓ |
|---|---|---|---|---|
| | 1 | $3.55 \pm 0.16$ | 0.420 | $0.028 \pm 0.004$ |
| | 2 | $3.71 \pm 0.12$ | 0.522 | $0.037 \pm 0.004$ |
| P-Flow | 5 | $4.01 \pm 0.10$ | 0.549 | $0.067 \pm 0.004$ |
| | 10 | $4.08 \pm 0.10$ | 0.544 | $0.115 \pm 0.004$ |
| | 20 | $4.14 \pm 0.10$ | 0.540 | $0.210 \pm 0.005$ |

## A.2  Model Comparison on VCTK Dataset

Considering the training dataset LibriTTS and the test set of LibriSpeech as in-domain data, it appears important to demonstrate results for out-of-domain prompting data. Accordingly, we compare P-Flow and VALL-E samples using the VCTK dataset [39]. We generate samples using the same prompts and sentences that were used for the VCTK samples on the VALL-E demo page. We provide our generated samples on our demo page.

As in the main section, we use 10 ODE steps and a guidance scale of 1. We measure WER and SECS for these VCTK samples and calculate the average values for each metric. To measure WER, we utilize the same ASR model as in Section 4. For SECS measurement, in the interest of a fair comparison, unlike what we did in other sections of our paper, we measure the SECS between generated and ground truth samples, rather than using the re-synthesized or encoded and decoded samples. Due to long leading and trailing silences in VCTK prompts, we measure SECS with the reference speech trimmed at 20dB.

We provide objective metrics for each model on the VCTK dataset in Table 7. Table 7 demonstrates that P-Flow achieves better WER and SECS compared to VALL-E. These results are similar to the LibriSpeech test clean results in Section 4.2.

In addition, we collect comparative mean opinion scores (CMOS) and comparative speaker similarity mean opinion score (SMOS) for VCTK samples, using a -2 to +2 scale. More than 50 human raters evaluate all the generated samples given 3-second reference data. Table 6 shows that human raters prefer P-Flow over VALL-E in terms of sample quality including acoustic quality, prosody and human likeness, and speaker similarity. These results show that P-Flow can achieve performance similar to or better than VALL-E on in-domain and out-of-domain data, in addition to providing users with an inference latency of 0.1 seconds using 10 ODE steps.

Table 6: Zero-Shot Speech Synthesis Subjective Metrics for VCTK dataset. Preference scores $\geq 0$ indicate preference for P-Flow.

| Model | CMOS↑ | SMOS↑ |
|---|---|---|
| P-Flow vs Vall-E | $\mathbf{0.188} \pm 0.10$ | $\mathbf{0.267} \pm 0.166$ |

Table 7: Objective metrics for zero-shot TTS evaluated on the VCTK dataset. We compare WER and SECS using all the VCTK samples from the VALL-E demo page.

| MODEL | WER↓ | SECS↑ |
|---|---|---|
| VALL-E | 4.3 | 0.452 |
| P-FLOW (PROPOSED) | **2.4** | **0.465** |

### A.3 Analysis on ODE sampling methods

We measure the pronunciation accuracy (WER) and speaker similarity (SECS) for two second-order ODE methods: midpoint and Heun's method, detailed in [18]. We evaluate these two second-order methods for sampling steps $N$, which have a similar number of function evaluations as 10 Euler steps. Table 8 shows that the midpoint method with $N = 4$ produces WER= 2.7 and SECS= 0.540, not better than the Euler method. On the other hand, Heun's method improves upon Euler's method with a similar number of function evaluations as described in [18].

Table 8: Objective metrics for P-Flow with different ODE sampling methods.

| MODEL | WER↓ | SECS↑ |
|---|---|---|
| P-FLOW (EULER METHOD, $N = 10$) | 2.6 | 0.544 |
| P-FLOW (HEUN'S METHOD, $N = 4$) | 2.6 | **0.552** |
| P-FLOW (MIDPOINT METHOD, $N = 4$) | 2.7 | 0.540 |

### A.4 Zero-shot TTS with Emotional Reference Speech

We provide generated samples using emotional reference samples, where each sample exhibits distinct prosody, as demonstrated in [37]. We extract reference speech samples from EmoV-DB [1], representing five different emotions. From each reference speech, we utilize a 3-second segment to perform zero-shot TTS. On our demo page, we present generated samples for the same sentence given the speech prompts for these five emotions. P-Flow, similar to VALL-E, utilizes a speech-prompted text encoder composed of an autoregressive transformer, enabling the generation of samples with different prosody based on the reference speech.

## B  Model Architectures

We provide explanations for each module in this section and detailed hyperparameters and architecture of P-Flow are shown in Table 9.

**Speech-prompted Text Encoder** Our text-encoder consists of several linear projection layers, a pre-network with 3 convolutional layers, and a 6-layer transformer with 2 attention heads of 192 hidden dimensions. The input to the text encoder is the speech prompt and text embeddings projected into the same dimensions. For the input of the speech-prompted text encoder, we project the speech prompt and text embeddings into the same dimension and input to the same pre-network. The resulting representation is then split into prompt and text parts, to which positional encodings are added. We define each positional encoding as the sum of absolute positional encoding and a learnable fixed-size embedding so that the transformer can differentiate the speech prompt and text through learnable embeddings. The representations of the speech prompt and text are then fed into a transformer architecture that allows each text position to attend to the speech prompt.

**Duration predictor** Our duration predictor is a shallow convolution-based model used in [21]. Since our text encoder output already provides speaker-conditional hidden representation, we use the hidden representation before linear projection to $h_c$ as the input of the duration predictor.

**Flow matching Decoder** Our flow matching decoder utilizes 18 layers of WaveNet-like architecture [36] with 512 hidden dimensions. We use the global conditioning method in WaveNet for conditioning $t$ and concatenate the aligned encoder output $h$ with the input $x_t$ along the channel axis for conditioning the speaker-conditional text representation.

Table 9: Hyperparameters of P-Flow

|  | Hyperparameter | |
|---|---|---|
| Speech-prompted Text Encoder | Phoneme Embedding Dim | 192 |
| | PreNet Conv Layers | 3 |
| | PreNet Hidden Dim | 192 |
| | PreNet Kernel Size | 5 |
| | PreNet Dropout | 0.5 |
| | Transformer Layers | 6 |
| | Transformer Hidden Dim | 192 |
| | Transformer Feed-forward Hidden Dim | 768 |
| | Transformer Attention Heads | 2 |
| | Transformer Dropout | 0.1 |
| | Prompt Embedding Dim | 192 |
| | Number of Parameters | 3.37M |
| Duration Predictor | Conv Layers | 3 |
| | Conv Hidden Dim | 256 |
| | LayerNorm Layers | 2 |
| | Dropout | 0.1 |
| | Number of Parameters | 0.36M |
| Flow Matching Decoder | WaveNet Residual Channel Size | 512 |
| | WaveNet Residual Blocks | 18 |
| | WaveNet Dilated Layers | 3 |
| | WaveNet Dilation Rate | 2 |
| | Number of Parameters | 40.68M |

