# OpenReview forum: "P-Flow: A Fast and Data-Efficient Zero-Shot TTS through Speech Prompting"
_NeurIPS.cc/2023/Conference — NeurIPS 2023 poster_

### Official Review · Reviewer_KPCd · 2023-07-06

**Soundness:** 3 good
**Presentation:** 3 good
**Contribution:** 3 good
**Rating:** 5
**Confidence:** 4

**Summary:**

The paper proposes P-Flow that can achieve high speaker similarity performance and fast inference speed on the zero-shot TTS Task. To improve the speaker similarity score, P-Flow uses  a speech prompted text encoder to generate speaker-conditional text representation for speaker adaptation. To achieve fast inference speed and high audio quality, P-Flow incorporates the flow matching generative decoder with the speaker-conditional text representation. Experimental results demonstrates that P-Flow trained on a relatively small dataset matches the speaker similarity performance of the large-scale zero-shot TTS models and has more than 20× faster sampling speed.

**Strengths:**

1. The proposed speech prompt approach for the non-autoregressive zero-shot TTS model is interesting, which surpasses the speaker embedding approach and provides in-context learning capabilities for speaker adaptation.
2. In experiments, P-Flow shows comparable speaker adaptation performance to the large-scale autoregressive baseline using significantly fewer training data and a small transformer-based encoder.
3. The flow matching generative decoder enjoys a faster sampling speed than previous neural codec language models while maintaining a good audio quality.

**Weaknesses:**

After listening to the audio samples in the demo page, my main concerns are mainly related to the unstable timbre issue. Please refer to the following Questions section for details.

**Questions:**

1. The audio samples provided by the authors on the demo page exhibit unstable timbre issues. For example, in ''Prompting v.s. Separate Speaker Encoder'', the timbre of the example 4 of P-Flow is unstable. Moreover, in ''Model Comparison'', the timbre of the example 1 of P-Flow is also unstable (e.g., ''...had fulfilled his mission.''). The authors should explain about this phenomenon.
2. The paper claims that P-Flow ''matches the speaker similarity performance of the large-scale zero-shot TTS models with two orders of magnitude less training data''. But the unstable timbre issue in Point 1 may due to the limited training data. I think the generalization capacity of the model should be further evaluated. For example, the LibriSpeech test-clean dataset (test set) and the LibriTTS dataset (training set) are in the same domain (both derived from read audiobooks from the LibriVox project). Further experiments on the VCTK or other datasets should be conducted to validate the generalization capacity of P-Flow with out-of-domain data.
3. In Section 4.1, the authors compare the proposed prompt-based speaker conditioning mechanism with the baseline that encodes all the speaker information into a fixed-size vector. After listening to the audio samples in the demo page, I believe the overall objective speaker similarity is improved by the proposed speech prompting approach. But the unstable timbre issue in Point 1 may also greatly influence the subjective results. The authors should conduct subjective evaluations for the ''Prompting v.s. Separate Speaker Encoder'' experiment to make it more convincing.
4. In Section 1, the paper claims that large scale datasets of tens of thousands of hours of audio data has not been convincingly proven to be necessary. I agree that P-Flow achieves similar results with VALL-E (a well-known large-scale TTS model). However, can the speaker similarity score of P-Flow be further improved by being trained on a relatively larger datasets? More experiments should be conducted to validate it.

**Limitations:**

Yes, the authors have addressed some of the impacts and limitations in Section 5.

---

> ### Author Rebuttal · Authors · 2023-08-09
>
> **Q1,3. Regarding timbre instability**
>
> After careful analysis, we believe that the event observed by the reviewer is a change in the fundamental frequency range. While timbre is defined to be F0 invariant [1], it is possible that a change in fundamental frequency range can be perceived as a change in timbre. Furthermore, given the nature of the data used to train the model (LibriTTS, narration of books), it is possible that the voice actors modify their voices to represent different characters in the story and, as such, it is possible that the model will replicate this behavior. Given our recent CMOS and SMOS results on VCTK and earlier results on LibriSpeech's test set, we believe that these events are either rare or marginally significant for human raters, given their consistent preference for P-Flow over VALL-E.
>
> **Q2. Generalization capacity of P-Flow with OOD data**
>
> As pointed out by Reviewer Bg7K and KPCd, considering the training dataset LibriTTS and LibriSpeech's test set as in-domain data, it appears important to demonstrate results for out-of-domain prompting data. Accordingly, we compared P-Flow and VALL-E samples using the VCTK dataset. We generated samples using the same prompts and sentences that were used for the VCTK samples on the VALL-E demo page. We have also provided our generated samples in the “Rebuttal 1” section of our demo page. We kindly ask the reviewers to check our VCTK experiments in the updated demo page.
>
> We generated samples using 10 ODE steps and a guidance scale of 1. We measured WER and SECS for these VCTK samples and calculated the average values for each metric. To measure WER, we utilized the same ASR model as in our paper. For SECS measurement, in the interest of a fair comparison, unlike what we did in other sections of our paper, we measured the SECS between generated and ground truth samples, rather than using the re-synthesized or encodec and decoded samples. Due to long leading and trailing silences in VCTK prompts, we additionally measured SECS with prompts trimmed at 20dB.
>
> The results below provides objective metrics for this experiment on the VCTK dataset and show that P-Flow achieves better WER and SECS similar to VALL-E when leading silence included. When leading silence is trimmed, P-Flow exhibits better SECS than VALL-E. These results are similar to the LibriSpeech test clean results in Section 4.2.
>
> **Table. Model comparison on VCTK dataset (Objective evaluation)**
>
> | Model        | WER| SECS (w/o trimming)                | SECS (w/ trimming prompt)  |
> |--------------|:--------------------------------------------:|:-------------------------------:|-------:|
> | VALL-E | 4.3|0.446 | 0.452 |
> | P-Flow | 2.4|0.441 | 0.465 |
>
> In addition, we collected Comparative Mean Opinion Scores (CMOS, ~ 50 human raters) and comparative Speaker similarity Mean Opinion Score (SMOS, ~ 100 human raters) for the above mentioned out-of-distribution samples, using a -2 to +2 scale. The results below show that human raters prefer P-Flow over VALL-E in terms of sample quality (acoustic fidelity, prosody and human-likeness) and speaker similarity.
>
> **Table. Model comparison on VCTK dataset (Subjective evaluation).**
>
> | Model        | CMOS $\uparrow$| SMOS $\uparrow$ |
> |--------------|:--------------------------------------------:|:-------------------------------:|
> | P-Flow vs. VALL-E | 0.188 $\pm$ 0.10|0.267 $\pm$ 0.166 |
>
> These results show that P-Flow can achieve performance similar or better than VALL-E on in-domain and out-of-domain data, in addition to providing users with an inference latency of 0.1 seconds using 10 ODE steps.
>
> **Q4. P-Flow with More Data**
>
> Our claim that large-scale datasets are unnecessary was due to poor choice in phrasing, which we thank the reviewer for pointing out. Our intent was only to demonstrate improved data efficiency, and not that large-scale datasets provide no benefit. We will rectify our phrasing in the draft.
>
> To better support our data efficiency claim, we fine-tuned P-Flow on 5K hours of data for 100K iterations, one order of magnitude more data than our original experiments, and computed WER and SECS on LibriSpeech-Test-Clean. Our results below are very promising, and suggest that with more data we can achieve even better speaker similarity while retaining good pronunciation accuracy (WER). We believe this is a promising direction to further explore, and expect that larger datasets of around 60K hours, similar in scale to what is used in VALL-E, could further improve P-Flow's speaker similarity score.
>
> **Table. Objective metrics for P-Flow trained with LibriTTS and fine-tuned with MLS 5K hours of data**
>
> | Model      | WER                | SECS |
> |--------------|:-------------------------------:|-------:|
> | P-Flow | 2.6 | 0.544 |
> | P-Flow (fine-tuned w/ MLS 5K hours of data, 100K iter) | 2.6 | 0.553 |
>
> [1] https://uen.pressbooks.pub/auralskills/chapter/timbre/
>
> [2] https://www.microsoft.com/en-us/research/project/vall-e-x/vall-e/

---

> > ### Comment · Area_Chair_reTd · 2023-08-18
> >
> > Thank you for providing the results of P-Flow trained with more data, the comparison between P-Flow and VALL-E, and answers to other questions.

---

### Official Review · Reviewer_zUdt · 2023-07-06

**Soundness:** 3 good
**Presentation:** 3 good
**Contribution:** 2 fair
**Rating:** 5
**Confidence:** 4

**Summary:**

This work propose a flow-matching zero-shot TTS model called P-Flow. It is fast and data-efficient in comparison with Vall-E.

**Strengths:**

I like the idea of flow matching, and it seems a new fashion for generative tasks. I do believe flow-matching will benefit the speech generation community.
The idea of prompting speaker encoder is novel and potentially effective.
I believe the clarity and quality of the presentation are good and easy to follow.
The samples in the demo page are promising.
The major claims are supported with in-depth ablation studies.


**Weaknesses:**

The overall novelty is limited. The main architecture of this model is very similar to recent NAR TTS models such as Glow-TTS, Grad-TTS. For example, all these models leverages MAS to compute alignments, and all these models have a powerful probabilistic decoder that
allows to learn complex distribution from known latent prior, such as normalizing flow (Glow-TTS), DDPM (Grad-TTS), and flow matching (P-Flow).

From my side, the experimentation is far from sufficient. As I said, this work is very similar to NAR TTS models like Glow-TTS and Grad-TTS. A direct comparison of P-Flow with Glow-TTS and Grad-TTS in the high quality TTS synthesis is essential, or the authors could report
the comparison of P-Flow with the promoting version of Glow-TTS or Grad-TTS in zero-shot scenarios.

The vocoder is trained separately.

**Questions:**

No

---

> ### Author Rebuttal · Authors · 2023-08-09
>
> We agree that our base model architecture is similar to existing Non-AutoRegressive (NAR) TTS methods (Glow-TTS and Grad-TTS). However, we believe our work provides valuable insight into the current state of zero-shot TTS research. The recently proposed language modeling-based zero-shot TTS model, VALL-E, achieved a breakthrough in the zero-shot TTS task, surpassing existing NAR zero-shot TTS models by a significant margin. Our work draws into question whether certain design choices in current SOTA models are truly necessary, and hopefully provides some guidance on promising directions of future research in this topic. Ideally, we would like to maintain the computational efficiency of NAR TTS frameworks while achieving zero-shot TTS capabilities at the lever of large autoregressive models such as VALL-E.
>
> We consider the reliance on a fixed-size speaker embedding vector for speaker adaptation in these NAR zero-shot TTS models as a factor that hinders zero-shot speaker adaptation compared to VALL-E's speech prompting method. To achieve our goal, we closely followed the base components of existing NAR TTS models and introduced speech prompting-based conditioning for zero-shot speaker adaptation. By introducing speech prompting-based speaker adaptation, we significantly enhance speaker similarity of NAR zero-shot TTS to a level compared to VALL-E, while retaining NAR models' faster inference speeds. Furthermore, our model has the advantage of not requiring a pre-trained speaker encoder, unlike the existing NAR zero-shot TTS models.
>
> Reviewer zUdt makes a reasonable request about providing a comparison to Glow-TTS or Grad-TTS. While we did not perform a zero-shot TTS comparison, **we provide a DDPM vs Flow Matching ablation on our demo page (Rebuttal 2)**, also detailed in our response to reviewer Bg7K. This section includes uploaded samples from two models to ablate our choice of using flow-matching. The model utilizing the DDPM decoder is akin to Grad-TTS, wherein a zero mean prior is employed to define forward and reverse processes. The second model utilizes the Flow matching decoder, and can be seen as a variant of P-Flow without speech prompt and masking. We sample from these models with varying amounts of sampling steps. Consistent with the results from the Grad-TTS demo page, the DDPM's decoder sample quality degrades severely with fewer than 50 steps. In contrast, the flow matching decoder produces high-quality samples with as low as 10 ODE sampling steps. Moreover, we observed that even with very few sampling steps, such as 2 or 4, this model generates reasonably good quality speech.
>
> We did not find it necessary to compare directly against these models because YourTTS [1], based on the superior VITS model already significantly lags behind VALL-E regarding speaker adaptation performance. Further, the extension of Glow-TTS to zero-shot TTS, SC-GlowTTS [2], also shows inferior zero-shot speaker adaptation performance compared to YourTTS.
>
>
> [1] Casanova et al., "YourTTS: Towards Zero-Shot Multi-Speaker TTS and Zero-Shot Voice Conversion for everyone"
>
> [2] Casanova et al., "SC-GlowTTS: an Efficient Zero-Shot Multi-Speaker Text-To-Speech Model"

---

> > ### Comment · Reviewer_zUdt · 2023-08-17
> > **Raise my score**
> >
> > Thanks for addressing my comments. The new ablation is convincing. Therefore,  I'd like to raise my score to 5: Borderline accept

---

> > > ### Author Response · Authors · 2023-08-17
> > >
> > > Thank you for responding to the rebuttal, and we're glad that the concerns were addressed. If it's alright, it would be great to directly incorporate the results from the rebuttal into the original rating.

---

### Official Review · Reviewer_Bg7K · 2023-07-06

**Soundness:** 2 fair
**Presentation:** 3 good
**Contribution:** 2 fair
**Rating:** 6
**Confidence:** 4

**Summary:**

The paper introduces P-Flow, a novel zero-shot text-to-speech (TTS) model that addresses the limitations of existing large-scale neural codec language models. P-Flow utilizes speech prompts for speaker adaptation and consists of a speech-prompted text encoder and a flow-matching generative decoder. The text encoder generates a speaker-conditional text representation using speech prompts and text input, while the generative decoder synthesizes high-quality personalized speech at a significantly faster speed than real-time. P-Flow is trained on the LibriTTS dataset using continuous mel-representation and achieves comparable speaker similarity performance to large-scale zero-shot TTS models but with significantly less training data and more than 20 times faster sampling speed. The results demonstrate that P-Flow offers improved pronunciation and is preferred in terms of human likeness and speaker similarity compared to state-of-the-art alternatives.

**Strengths:**

- Novel approach: P-Flow introduces a unique methodology for zero-shot TTS that combines speech prompts and text input for speaker adaptation, resulting in improved speech synthesis quality and faster synthesis speed.
- Data efficiency: P-Flow achieves comparable speaker similarity performance to large-scale models but with only a fraction of the training data, making it more data-efficient.
- Improved sampling speed: P-Flow demonstrates a significant improvement in sampling speed compared to previous autoregressive TTS methods and large-scale neural codec language models, enabling real-time or near-real-time synthesis.
- High-quality synthesis: The flow-matching generative decoder in P-Flow produces high-quality personalized speech, enhancing the naturalness and human-likeness of the synthesized audio.
- Preference over state-of-the-art: P-Flow is preferred over recent state-of-the-art counterparts in terms of pronunciation, human likeness, and speaker similarity, making it an attractive alternative for zero-shot TTS.

**Weaknesses:**

- The motivation of using the flow-matching-based decoder is not very clear. This mask and prediction training schema can be applied to other generative models such as GAN and DDPM/DDIM. What makes you believe that the flow-matching model is the most appropriate choice?
- The evaluation of zero-shot performance in real scenarios is not verified. Although this paper assures no overlap exists with our training data, as Libri-TTS (the training set) is derived from LibriSpeech (the test set), the current results are not yet sufficiently convincing to demonstrate its ability and performance in real zero-shot scenarios.

**Questions:**

- Can you provide visualizations of the generated mel-spectrograms to assist us in assessing the quality of the synthesized speech?
- Why do you prefer using the flow-matching model as the decoder instead of diffusion models like DDPM? What advantages does this model offer in zero-shot TTS tasks? If you're looking to speed up inference, there are also numerous acceleration methods available for diffusion models.
- Libri-TTS is derived from LibriSpeech and can be considered as in-domain data. Can you provide results using out-of-domain prompting data, such as speech audio from podcast or YouTube, to assess the zero-shot cloning performance in real scenarios?

**Limitations:**

The authors have adequately addressed the limitations.

---

> ### Author Rebuttal · Authors · 2023-08-09
>
> **Motivation for using a flow matching decoder**
>
> The precursor papers on conditional flow matching [1, 2] provided evidence of the advantages of flow matching versus DDPM, especially flow matching's ability to simulate simpler trajectories while retaining good sample quality with fewer ODE steps than DDPM.
>
> Thus, in the initial phase of our experiments, we evaluated sample quality across various sampling steps using flow matching and DDPM decoders on a single-speaker TTS dataset, LJSpeech. Both models consist of the same text encoder (6-layer Transformer), alignment search algorithm (MAS), and decoder (18-layer WaveNet). The model using the DDPM decoder was trained for 400K iterations with an encoder loss + score matching loss, while the model using the flow matching decoder was trained for 400K iterations with encoder loss + flow matching loss. Additionally, we evaluated both models using Euler ODE sampling without classifier-free guidance. In our initial experiments, we did not use speech prompting and masking in P-Flow.
>
> We uploaded to our demo page (Rebuttal 2 Section) samples from two models to ablate and support our choice in using flow-matching.  The model utilizing the DDPM decoder is akin to Grad-TTS, wherein a zero mean prior is employed to define forward and reverse processes. The second model utilizes the flow matching decoder, and can be seen as a variant of P-Flow without speech prompt and masking. We sample from these models with varying amounts of sampling steps. Consistent with the results from the Grad-TTS demo page, the DDPM's decoder sample quality degrades severely with fewer than 50 steps. In contrast, the flow matching decoder produces high-quality samples with as low as 10 ODE sampling steps. Moreover, we observed that even with very few sampling steps, such as 2 or 4, this model generates reasonably good quality speech.
>
> Given these results, we decided to develop on top of the strong flow matching baseline instead of finding mechanisms, such as distillation, to improve the DDPM decoder such that it meets the flow matching baseline. Because these initial experiments do not directly relate to zero-shot speaker adaptation, we chose not to include them in the paper at the time of submission. We will revise our draft to include them in the appendix.
>
> **Visualizations of the generated mel-spectrograms**
>
> As requested by Reviewer Bg7K, we present visualizations of mel-spectrograms generated by P-Flow in the official comment PDF file. These mel-spectrograms are generated using three sentences used for model comparison in the demo.
>
> **OOD prompting data**
>
> As pointed out by Reviewer Bg7K and KPCd, considering the training dataset LibriTTS and LibriSpeech's test set as in-domain data, it appears important to demonstrate results for out-of-domain prompting data. Accordingly, we compared P-Flow and VALL-E samples using the VCTK dataset. We generated samples using the same prompts and sentences that were used for the VCTK samples on the VALL-E demo page. We have also provided our generated samples in the “Rebuttal 1” section of our demo page. We kindly ask the reviewers to check our VCTK experiments in the updated demo page.
>
> We generated samples using 10 ODE steps and a guidance scale of 1. We measured WER and SECS for these VCTK samples and calculated the average values for each metric. To measure WER, we utilized the same ASR model as in our paper. For SECS measurement, in the interest of a fair comparison, unlike what we did in other sections of our paper, we measured the SECS between generated and ground truth samples, rather than using the re-synthesized or encodec and decoded samples. Due to long leading and trailing silences in VCTK prompts, we additionally measured SECS with prompts trimmed at 20dB.
>
> The results below provides objective metrics for this experiment on the VCTK dataset and show that P-Flow achieves better WER and SECS similar to VALL-E when leading silence included. When leading silence is trimmed, P-Flow exhibits better SECS than VALL-E. These results are similar to the LibriSpeech test clean results in Section 4.2.
>
> **Table. Model comparison on VCTK dataset (Objective evaluation)**
>
> | Model        | WER| SECS (w/o trimming)                | SECS (w/ trimming prompt)  |
> |--------------|:--------------------------------------------:|:-------------------------------:|-------:|
> | VALL-E | 4.3|0.446 | 0.452 |
> | P-Flow | 2.4|0.441 | 0.465 |
>
> In addition, we collected Comparative Mean Opinion Scores (CMOS, ~ 50 human raters) and comparative Speaker similarity Mean Opinion Score (SMOS, ~ 100 human raters) for the above mentioned out-of-distribution samples, using a -2 to +2 scale. The results below show that human raters prefer P-Flow over VALL-E in terms of sample quality (acoustic fidelity, prosody and human-likeness) and speaker similarity.
>
> **Table. Model comparison on VCTK dataset (Subjective evaluation).**
>
> | Model        | CMOS $\uparrow$| SMOS $\uparrow$ |
> |--------------|:--------------------------------------------:|:-------------------------------:|
> | P-Flow vs. VALL-E | 0.188 $\pm$ 0.10|0.267 $\pm$ 0.166 |
>
> These results show that P-Flow can achieve performance similar or better than VALL-E on in-domain and out-of-domain data, in addition to providing users with an inference latency of 0.1 seconds using 10 ODE steps.
>
> Additionally, the results we presented for emotional reference speech in the demo page and Appendix 2.2 section were generated using out-of-domain data, the EmoV-DB dataset.
>
> [1] Lipman et al., "Flow Matching for Generative Modeling"
>
> [2] Tong et al., "Improving and generalizing flow-based generative models with minibatch optimal transport"

---

> > ### Comment · Area_Chair_reTd · 2023-08-18
> >
> > Thank you for providing the comparison results for VALL-E and P-Flow in terms of both objective and subjective evaluations, the mel-spectrogram generated by P-Flow, and answers to other questions.

---

### Official Review · Reviewer_9LwJ · 2023-07-24

**Soundness:** 4 excellent
**Presentation:** 3 good
**Contribution:** 3 good
**Rating:** 6
**Confidence:** 4

**Summary:**

The paper proposes a zero-shot TTS model, P-Flow, which combines a speech-prompted text encoder and a flow-matching generative decoder. The encoder generates a speaker-conditional text representation using the target speech prompt and text, while the decoder utilizes conditional flow matching to model the conditional distribution of mel-spec.  The evaluation shows that P-Flow achieves comparable speaker adaptation performance to the large-scale autoregressive models (e.g., VALL-E) with significantly fewer training data and more than 20x faster sampling speed. Subjective testing also shows human listeners prefer the audios generated by P-Flow compared to VALL-E due to its better pronunciation accuracy and naturalness.

**Strengths:**

1. The paper challenges the recent trend that using LM-similar training approach for speech synthesis, instead, it argues that non-autoregressive formulation using the cheaper traditional representations such as mel-spectrograms can make speech synthesis both fast and high-quality.
2. The paper proposed a novel speech-prompted text encoder. Combining the text and speech prompt, this encoder tries extracting speaker information from the prompt and generating a speaker-conditional text representation. Specifically, the MAS algorithm is utilized to align the text encoder output with the mel-spectrogram, and the training target of the encoder is set to minimize the distance between the text encoder representation and the mel-spec. It not only distills speaker information to the encoder representation but also provides a way for end-to-end training for the whole system.
3. The paper argues that it is sufficient to use the first-order Euler's method to solve the ODE during inference, instead of depending on the off-the-shelf numerical ODE solvers.
4. The paper proposes to guide the sampling trajectory away from the average feature vector computed from h.
5. The paper explains well why masked loss is necessary, i.e., preventing the model from collapsing to a trivial copy-pasting. Other works such as voicebox also utilizes masked loss but they do not provide a good explanation.
6. The paper conducts extensive experiments to evaluate the performance of P-Flow in terms of objective metrics (WER, SECS, inference latency) and subjective metrics (CMOS, SMOS). The paper also provides an ablation study to show the effectiveness of speech prompting and guidance methods.
7. Both the experiment results and demo audios show that P-Flow achieves lower WER, competitive SECS, much smaller inference latency, and much higher human preference scores, compared with VALL-E.

**Weaknesses:**

1. LM-similar training approach, i.e., modeling the sequential discrete tokens in autogressive way, is supper good at long sequences, which has been demonstrated by the success of ChatGPT with very long input tokens. Since the paper is trying to argue that using non-autoregressive method with mel-spectrum representation is at least as good as the modeling metrology proposed in VALL-E, a very important experiment is to show the model's performance in terms of the very long prompt text, which is missing from this paper.
2. The paper shows that P-Flow generated audios are preferred by human listeners compared to VALL-E, and it also points out that the difference is not only attributed to the model but also the performance degradation introduced by the neural codec. A very important experiment is to show compared to Encodec, what is performance gain can be achieved by the hifi-gan vocoder. This can help the readers to get a better insight regarding the potential of the model itself.
3. duration predictor is very important in P-Flow, however the paper does not provide the experiment results of the duration predictor, at least it should be included in the supplemental materials.
4. The paper uses the first-order Euler's method to solve the ODE during interference. How this simple inference comparing to the off-the-shelf ODE solvers?
4. There are still some important things which are not clarified well.
   1) \hat_{h} is not well explained, how to calculate \hat_{h}?
   2) In table 1, the GT achieves higher WER comparing to P-Flow, why?
   3) during inference, \hat_{d} is used to expand h_c to h. How about training, do you use d or \hat_d?
   4) In table 2, the inference latency of VALL-E is approximated using the size of the transformer. The readers cannot tell how accurate the approximation is and I would suggest the authors to give more details regarding the approximation in the supplemental materials.
5. There are some typos: Line 147, "it also serves encourages the" should be "it also encourages"; L176, "objective (L_{dur})is to minimized" should be "objective (L_{dur}) is to minimize"

**Questions:**

1. how to calculate \hat_{h} in the guided sampling? If it is training dataset dependent, how well it can generalize?
2. During training, is d or \hat_d used to expand h_c to h?
3. How is the model performance with long text prompt?  e.g., 100s of words
4. How duration model itself performs? Is it a bottleneck for further improving the model's performance?
5. Compared to neural codec (Encodec), how much gain the hifi-gan vocoder itself can achieve?
6. How about other ODE-solving algorithms? Is the current solution achieves the best trade-off between time cost and accuracy?

**Limitations:**

1. The paper discussed a little bit regarding the potential ethical and social implications of P-Flow, such as voice cloning and impersonation. I would suggest the author either develop another model to identify whether one audio is generated by P-Flow or use existing open-sourced fake audio detector to evaluate whether the synthesised audio can be identified from the real audios.

---

> ### Author Rebuttal · Authors · 2023-08-10
>
> **W1 and Q3 - P-Flow with long prompt**
>
> Concerning P-Flow for long prompts, we began by taking inspiration from the VALL-E and SPEAR-TTS, which demonstrated effective voice cloning using only 3 seconds of voice data. Based on this, we fixed our speech prompt length at 3 seconds for all experiments. Our focus was on showing that non-autoregressive models can achieve similar levels of speaker adaptation using the 3-second prompt, as shown in the previous LM-similar approach. Exploring speaker adaptation using longer prompts would require training the model to perform generative modeling on variable-length segments masked during the training process. We leave this approach as future work.
>
> **W2 and Q5 - Gain achieved by using HiFi-GAN**
>
> We believe that properly ablating on the effect of the vocoder or neural condec in this context is a very challenging task: on the one hand it would require adapting P-Flow, a flow-matching model operating on continuous data, to operate on discrete EnCodec codes; on the other hand, it would require adapting VALL-E, a Autoregressive Transformer operating on discrete codes, to operate on continuous data. We hope the reviewer would agree that each of these tasks is a research project on its own that would go beyond the scope of this rebuttal.
> Nonetheless, we agree with reviewer 9LwJ that the choice of neural codec or vocoder is an important design choice. As such, we provide an upperbound in SECS by computing it between different ground truth audio and samples from the same speaker re-synthesized with EnCodec and HiFiGAN:
>
> - SECS (GT 3-sec prompt, GT) = 0.669
> - SECS (GT-EnCodec 3-sec prompt, GT-EnCodec) = 0.622
> - SECS (GT-HiFiGAN 3-sec prompt, GT-HiFiGAN) = 0.643
>
> In the case of using EnCodec, the SECS results obtained by resynthesizing with GT are lower compared to HiFi-GAN. This implies that P-Flow gains an advantage by using mel-spectrograms and HiFi-GAN instead of EnCodec.
>
> **W3 and Q4 - Duration predictor**
>
> P-Flow employs a deterministic duration predictor consisting of shallow convolutional layers. It takes the speaker-conditional output from the text encoder to regress phoneme duration values. We conducted model comparisons using the speaker-conditional durations predicted by the duration predictor. The results from CMOS or SMOS, among others, confirm the duration predictor's capability to generate reasonable phoneme durations.
>
> To demonstrate the performance bound of the duration predictor itself, we employed ground truth audio, similar to the training process, to derive the optimal duration $d$ using the Monotonic Alignment Search (MAS) algorithm. By using this optimal duration $d$ during inference, we established an upper bound for SECS using the same encoder and decoder parameters as P-Flow. When employing the optimal duration, the upper bound SECS value was determined to be 0.572 using 10 Euler steps and guidance scale 1. This finding indicates that by adjusting durations alone, a significant improvement in speaker similarity can be achieved, compared to the 0.544 SECS score for P-Flow in Table 2 of the paper.
>
> In an attempt to enhance duration prediction, we explored training a transformer-based duration predictor using regression loss. However, our efforts did not result in meaningful improvement over the 0.544 SECS score. We noted that straightforward architectural modifications to the duration predictor within P-Flow did not yield substantial improvements in duration prediction performance. As indicated in the conclusion, further advancements in duration prediction are reserved as a topic for future work in this paper.
>
> **W4 and Q6 - Different ODE samplers**
>
> We reported only the results obtained using a simple first-order Euler method with a small number of sampling steps, as it provided favorable outcomes. As suggested by Reviewer 9LwJ, showing the results using different ODE sampling methods would greatly benefit the paper. This could have led to achieving an even better SECS, and we will incorporate these results in the appendix.
>
> We measure the pronunciation accuracy (WER) and speaker similarity (SECS) for two second-order ODE methods: midpoint and Heun's method. We evaluated these two second-order methods for sampling steps N, which have a similar number of function evaluations as 10 Euler steps.
>
> The midpoint method with N=4 produces WER=2.7 and SECS=0.540, not better than the Euler method. On the other hand, Heun's method improves upon Euler method:
>
> **Table. Objective metrics for P-Flow with different ODE sampling methods.**
>
> | Model        | WER| SECS |
> |---|:----:|:---:|
> | P-Flow (1st-order Euler method, N=10) | 2.6|0.544 |
> | P-Flow (2nd-order Heun's method, N=4) | 2.6|**0.552**|
> | P-Flow (2nd-order midpoint method, N=4) | 2.7|0.540|
>
> **W5 and Q1, 2 - Missing details**
>
> - $\bar{h}$ is a representation that involves averaging the expanded representation $h$ along the time axis to obtain a fixed-size vector, which is then duplicated along the time axis. This is not a pre-computed value derived from the training dataset; rather, it is a representation computed during the inference process by averaging and duplicating the expanded representation $h$.
>
> - We apologize for the confusion. It was our mistake. WER and SECS values for GT in Table 1 must be the values in Table 2: WER=2.4 and SECS=0.64.
>
> - As in line 153, during training, we use the duration $d$ obtained by MAS to expand $h_c$ to $h$.
>
> - We will provide specific details about measuring the inference latency of VALL-E in the appendix. Following VALL-E's paper, we measure inference on a 12-layer transformer with a hidden size of 1024 and 16 heads. Given that VALL-E's neural audio codec representation is at 75Hz, we approximate the inference latency of VALL-E by measuring the latency required for 75*n + 7 forward passes to generate $n$ seconds of audio.
>
> **W6 - typos**:
> Will be corrected

---

> > ### Comment · Area_Chair_reTd · 2023-08-18
> >
> > Thank you for providing detailed answers to the questions and for clarifying the information.

---

### Author Rebuttal · Authors · 2023-08-10

Thank you for taking the valuable time to review our paper and providing helpful feedback. All the questions will greatly contribute to improving our paper.

We have provided responses to the weaknesses and questions mentioned by each reviewer. Additionally, we have included figures and tables for additional results in the attached PDF. In response to Reviewer Bg7K's request, we have added figures in the PDF showing the mel-spectrograms generated using P-Flow.

To summarize the tables presented in the PDF:
1. In response to Reviewer 9LwJ's suggestion about using a different ODE method instead of the Euler method for sampling, we conducted experiments as suggested and included the results in Table 1. We demonstrated that using the 2nd order Heun's method yields better objective metrics for speaker similarity compared to the Euler method.
2. Furthermore, as requested by Reviewers Bg7K and zUdt, we conducted experiments on out-of-domain data using LibriTTS and LibriSpeech. The results of these experiments are shown in Tables 2 and 3. We compared the performance of P-Flow and VALL-E on the out-of-domain VCTK dataset and demonstrated that P-Flow performs well even in an OOD domain it hasn't been trained on. The generated samples of VCTK using VALL-E and P-Flow have been uploaded to the existing demo page.
3. In response to Reviewer zUdt's query about the effects of using a larger dataset, we present the results in Table 4. We showed that fine-tuning the previously trained P-Flow on 5K hours of data can improve SECS.

Lastly, as Reviewer Bg7K asked for an explanation of the motivation behind the use of the flow matching decoder, we have uploaded additional sample results on the demo page. These results are from a single-speaker TTS task, comparing the flow matching decoder trained on LJSpeech with the DDPM decoder from the early stages of our research.

---

> ### Author Response · Authors · 2023-08-21
>
> We thank all the reviewers for taking the time to review our paper. We have addressed the comments from each reviewer, and one reviewer adjusted the score upwards following our rebuttal. We would like to respectfully remind everyone that the author-reviewer discussion period will soon conclude. We would greatly appreciate any further feedback on our rebuttal.

---

### Decision · Program_Chairs · 2023-09-21

**Decision:**

Accept (poster)

**Comment:**

This paper introduces P-Flow, a fast and efficient zero-shot Text-to-Speech (TTS) model that utilizes speech prompts for speaker adaptation. P-Flow is capable of achieving speech sampling that is 20 times faster. The authors’ rebuttals successfully addressed the concerns raised by the reviewers, resulting in three out of four reviewers raising their scores. I recommend that this paper be accepted.